# Investigating the Recyclability of Laser PP CP 75 Polypropylene Powder in Laser Powder Bed Fusion (L-PBF)

**DOI:** 10.3390/polym14051011

**Published:** 2022-03-03

**Authors:** Fredrick M. Mwania, Maina Maringa, Jacobus G. van der Walt

**Affiliations:** Department of Mechanical and Mechatronics Engineering, Central University of Technology, Private Bag X20539, Bloemfontein 9300, Free State, South Africa; mmaringa@cut.ac.za (M.M.); jgvdwalt@cut.ac.za (J.G.v.d.W.)

**Keywords:** recyclability, virgin material, aged powder, re-use cycle, melt flow index, thermal properties, dimensional error, Laser PP CP 75

## Abstract

In the present study, recyclability of Laser PP CP 75 polypropylene powder from Diamond Plastics GmbH was determined by characterizing and comparing the used powder after each cycle with material from previous cycles and with fresh powder. The melt flow index of Laser PP CP 75 was affected by recycling since it was observed to change by 30.62% after the 8th 100% re-use cycle, a lower value than PA 12 of 66.04%, for the 6th re-use cycle. Parts printed with virgin Laser PP CP 75 had an average dimensional error of 3.02% (virgin material) and 4.06% after the 4th 100% re-use cycle, which raises concerns about the commercial viability of the material. After the 4th re-use cycle, the printed parts had distorted edges and failed to print after the 9th print cycle. Lastly, tensile testing revealed a skewed bell-shaped curve of strength versus the number of recycles with the highest ultimate tensile strength occurring for the second 100% re-use cycle (7.4 MPa). The curves for elastic modulus and percentage elongation were inverted with minimum points for the 2nd 100% re-use cycle. Overall, the experimental work confirmed that the properties of polypropylene powder were affected by recycling in polymer laser sintering, but the powder exhibited superior characteristics upon recycling to those of the predominantly used PA 12 powder.

## 1. Introduction

Laser Powder bed fusion (L-PBF) is one of the most common additive manufacturing (AM) techniques for polymers. The process uses a heat source to fuse powder materials to create three-dimensional objects from model data by building components layer-upon-layer [1]. The polymer L-PBF strategy has considerable advantages over other AM technologies because it does not require support structures, since the unsintered powder supports the printed parts [1]. This advantage also allows for printing complex parts with high geometrical and dimensional accuracy and excellent mechanical properties, since post-processing is not needed. Furthermore, many parts can be stacked in the powder bed, increasing productivity [1]. The process can also be employed for different polymer materials, making it more popular. However, polymer materials are subjected to high temperatures for long hours during laser sintering, which degrades them, thus hindering subsequent recycling.

Available commercial polymers are limited because of stringent processing and material property requirements. Some commercially available polymers include PA 12, polyamide 11 (PA 11), polypropylene (PP), polyether ether ketone (PEEK), polystyrene (PS), such as Primecast 101 from EOS, and thermoplastic polyurethane (TPU) [2,3]. PA 12 and its blends are the most widely used polymers in L-PBF, making up about 95% of the use [1]. This is attributed to their excellent laser sintering processability, including almost spherical-shaped particles, a wide sintering window, and good powder flowability [4]. The limited array of commercial polymers for L-PBF has resulted in their high costs, hampers uptake of the technologies and increases the need to re-use the unfused powder [5]. According to Duddleston [5], one kilogram of PA 12 for L-PBF applications costs about USD 100. Besides, during the L-PBF processing of polymers, most of the material is not sintered; only 5–20% of the powder introduced into the process chamber is utilized [6,7]. Therefore, it is economical to re-use the unfused powder. The properties of aged PA 12 can be ameliorated by adding virgin material in specific ratios. Pilipović et al. [8] proposed a ratio of 67% of used powder to 33% of virgin PA 12 powder. The mixing ratio suggested by the manufacturers, EOS and 3D Systems, for PA 12 are 70:30 and 50:50 for aged to virgin powder, respectively [9]. Pilipović et al. [8] state that there is no mention of a possibility of re-using 100% of the polymeric materials. However, Diamond Plastics GmbH states that Laser PP CP 75 is 100% re-usable. In this regard, the present study investigates the recyclability of a new polypropylene material (Laser PP CP 75) from Diamond Plastics to determine if it is 100% re-usable. Laser PP CP 75 is an olefin polymer used in L-PBF of polymers to manufacture consumer products, such as kitchen utensils. The material is also meant for prototyping and manufacturing functional products to reduce the price of available commercial polymers. It is utilized to manufacture packaging films, boxes, bags, brushes, medical, and surgical equipment, and automotive parts [9].

Significant research has focused on the aging of polymers in the L-PBF process to improve the processability of used powders. Deterioration of polymers due to laser sintering can be defined as any change in either the molecular or phase structure of a material that alters its physical and chemical properties [5]. Physical aging affects the ordering of molecules, resulting in agglomeration, post-crystallization, and relaxation [10]. Post-crystallization is a physical change that increases the degree of crystallization of a polymer, while relaxation causes a reduction in stress in response to strain. Chemical degradation changes the structure of the material and is caused by oxidation, hydrolysis, and post-condensation [10]. Physical deterioration is reversible, unlike chemical aging, and both influence different aspects of polymers from storage manufacturing to service [5]. At high processing temperatures, dissolved oxygen and water environments cause polymer aging during L-PBF [11]. Heating polymers under oxygen-rich conditions results in the yellowing of the materials due to the formation of impurities [5]. In addition, the aging of polymers can be attributed to chain scission, chain branching, or cross-linking caused by high temperature [11]. These processes start with the formation of radicals at temperatures close to the melting point of the material in an auto-oxidation reaction [11]. The molecular weight of polymers, such as PA 12, increases after combining with radicals during post-condensation, thus altering the physical and chemical properties of the materials [11]. Overall, polymers degrade when subjected to L-PBF processing, which manifests in changed melting, crystallization, and flow properties of polymers and part properties, such as surface roughness and mechanical properties.

It is crucial to re-use unfused polymeric powder in L-PBF manufacturing to reduce the cost of the manufacturing process. However, the re-use of polymers in the L-PBF process is limited because it alters the chemical and physical properties of the materials. The technique is sensitive to different material properties required for successful processing. After recoating, an appropriate material should have adequate flow characteristics to achieve a smooth layer [4]. The powder particles should also be near-spherical to ensure suitable spreading. The material should also absorb sufficient laser energy for the complete coalescence of particles [4]. Moreover, too much heat should not be applied to cause thermal degradation of the powder during printing [4]. The material should have a wide sintering window to prevent rapid cooling, which encourages shrinkage and curling, thus affecting the physical parameters of printed components (surface roughness, dimensional and geometrical accuracy) [12,13]. The viscosity of the melt should be low enough for a complete fusion of particles but high enough to prevent the material from sinking into the powder surrounding the print, which might cause porosity, thus influencing the mechanical properties of the printed parts [4]. Therefore, it is imperative to determine the suitability of the unsintered polymer powder after every print cycle to determine if it can be re-used.

## 2. Literature Review

Extensive research has been directed towards the recyclability of polymers used in L-PBF. The results gathered proves that the L-PBF process alters both the physical and chemical properties of polymers. Chen et al. [14] investigated the recyclability of PA 6 powder in comparison with PA 12. The authors found that the morphology of both virgin powders is relatively spherical (Figure 1a,d). However, PA 6 breaks down into fragments after some degree of use, which exposes the flowability agents in the powder (Figure 1b,c). Hence, the morphology of PA 6 becomes rough and irregular after a single print cycle. It was found that PA 12 particles crack after exposure to sintering, resulting in the development of small round particles, as illustrated in Figure 1e,f. The authors also discovered that the particle size distribution of the two powders changes differently after re-use, as presented in Table 1. The particle size distribution of PA 12 does not change significantly, while PA 6 goes through considerable change, probably due to the breakdown phenomenon observed after exposure to heat. Lastly, the sintering window and crystallinity for PA 6 reduce notably, whereas changes for PA 12 are minute, as illustrated in Table 2. The findings confirm the conclusions by Yao et al. [15] that there is a slight difference of 4 μm in the powder shape, size, and distribution of recycled PA 12 powder.

Wudy & Drummer [16] investigated the molecular weight and thermal properties of aged PA 12. The authors found that the average molecular weight of the material increases with cumulative build time and build chamber temperature to a certain point and then starts to decrease, as illustrated in Figure 2 and Figure 3, respectively. An increase in molecular weight lowers the viscosity of a material, thus hampering recyclability because it reduces the fusion of melt particles, which undermines the mechanical strength of printed 3D parts. Furthermore, the melting temperature of PA 12 increases with re-use cycles, which might be attributed to post-sintering crystallization that results in the elongation of chains and, in turn, an increase in the molecular weight of a polymer. In conclusion, L-PBF degrades PA 12 material and causes unwanted “orange peel” formation, which affects the surface roughness of printed parts.

Ziegelmeier et al. [17] compared the aging behavior of thermoplastic polyurethane (TPU) with Duraform Flex (DF) from 3D Systems. The materials were re-used 14 times without refreshing. The results showed that the powders could be used without refreshing for 14 processing cycles, but changes in mechanical characteristics of the printed parts were evident. It was established that TPU has superior mechanical properties compared to DF. When recycled, both materials showed slight changes in the ultimate tensile strength, Young’s modulus, and percentage of elongation at break, as illustrated in Figure 4. Overall, the mechanical properties of components printed using TPU and DF changed slightly with re-use cycles.

Ziegelmeier et al. [17] also found that the melt viscosity of TPU and DF reduced with aging states, as illustrated in Figure 5. This phenomenon can be ascribed to reducing the backbone chain length due to random scission. They concluded that the processability of both TPU and DF was affected by re-use cycles.

## 3. Methodology

A set of test coupons at predetermined positions in the build volume (shown in Figure 6) of an EOSINT P 380 laser sintering AM machine from EOS, Germany was printed using Laser PP CP 75 powder from Diamond Plastics GmbH, Gräfenberg, Germany. The manufacturer’s parameters were used, except for the process and removal chamber temperatures, as illustrated in Table 3. The Laser PP CP 75 powder remaining in the machine and the cake powder surrounding the coupons were taken out of the AM machine, thoroughly mixed using a concrete mixer for 30 min, and a sample of the mixed powder was taken and kept for analysis. The powder was then reintroduced into the machine, another set of test coupons was produced in nine printing cycles, and the powder was re-used for eight cycles. The printing and cooling times for each cycle were about two hours, respectively. The building chamber was preheated for 120 min for each cycle. The batches of powder sampled at each cycle were assessed using scanning electron microscopy (SEM), differential scanning calorimetry (DSC), thermogravimetric analysis (TGA) analysis, and melt flow index (MFI) testing. The printed parts were subjected to physical inspection, tensile testing, and measurements to assess their dimensional accuracy.

A JEOL JSM-6610LV scanning electron microscope was utilized for SEM testing to determine the morphology of the particles. The samples were first coated with carbon to avoid supercharging, and the accelerating voltage was set to 30 kV. The SEM images were assessed using ImageJ software to determine the particle size distribution. DSC was conducted using the Mettler Toledo DSC apparatus. About six grams of powder was subjected to heating and cooling cycles performed between room temperature and 180 °C at a rate of 10 °C/min and a flow of 50 mL/min of nitrogen. Thermogravimetric analysis was conducted using a Mettler Toledo (TGA/SDTA851) TGA apparatus. The samples were heated at 10 °C/min from room temperature to 550 °C in a nitrogen environment. A tester from Ametex Ltd, Pretoria, South Africa. was utilized for MFI testing. About 6 g of powder was preheated for about 6 min at 230 °C (as recommended by the manufacturer), and a test load of 4.32 kg was used during testing. Lastly, the built specimens were tested using an MTS Criterion ^TM^, Model 43 universal testing machine, where ISO 572-2 standard was employed at an elongation rate of 1 mm/min. The dimensional accuracy of the printed parts was expressed using error *S1*, where ten measurements were taken for five samples and the average value was determined. The dimensions measured include the total length, width, and thickness of each specimen.

## 4. Results and Discussion

The reusability of the PP powder in this study was determined by characterizing and comparing the used powder with the fresh powder after each printing cycle (1st, 2nd, 3rd, 4th, 5th, 6th, 7th, 8th, and 9th print cycles). The printed specimens were physically examined for the presence of “orange peel,” which is common with PA 12. Scanning electron microscopy, DSC, TGA, and MFI testing were also used to characterize different powder batches. Lastly, uniaxial tensile testing and dimensional analysis tests were employed to describe the mechanical and physical properties of the printed specimens. The relationship between printing cycles and re-use cycles is as follows:
1—Printing cycle using virgin material (1st print cycle)—zero re-use cycle;2—2nd print cycle (1st re-use cycle);3—3rd print cycle (2nd re-use cycle);4—4th print cycle (3rd re-use cycle);5—5th print cycle (4th re-use cycle);6—6th print cycle (5th re-use cycle);7—7th print cycle (6th re-use cycle);8—8th print cycle (7th re-use cycle);9—9th print cycle (8th re-use cycle).


### 4.1. Physical Inspection

Figure 7 shows the parts printed for the 2nd, 3rd, 4th, 5th, 6th, and 9th print cycles, respectively. No “orange peel” was observed on the parts printed using re-used powder. The findings are in contrast to those of a study by Dotchev et al. [18], which found that when PA 12 from EOS is recycled five times with or without small quantities of new material; the parts produced have a rough surface finish due to the formation of an “orange peel”. The “orange peel” affects surface roughness, dimensional tolerance, and geometrical accuracy [6,7,18]. PP CP 75 has the upper hand over PA 12 regarding the “orange peel” phenomenon.

It was also found that the geometry of the tensile test specimens began to deteriorate after the first five printing cycles. Deformation of the edges became prominent on the 9th print cycle, and the recoater dislodged parts, causing the process to stop. Thus, it can be concluded that Laser PP CP 75 powder is re-usable for the first eight printing cycles without affecting the printing process. It was also evident that the geometry of the printed parts was significantly affected after the first five printing cycles.

### 4.2. Characterization of Powder to Establish Recyclability of Laser PP CP 75

Powder characterization of Laser PP CP 75 was undertaken using MFI testing, DSC, TGA, and SEM analysis to establish what changes took place after each print cycle. The MFI experiments determined the changes in powder flowability, while the DSC analysis identified the melting point of the powder and changes in the sintering window after each print cycle. The TGA analysis was conducted to investigate any changes in the degradation temperatures of the powder. Lastly, SEM testing determined the changes in powder and particle size morphology after each re-use cycle.

#### 4.2.1. Melt Flow Testing of Laser PP CP 75 after 100% Re-Use

Table 4 summarizes the MFI values for the nine batches of powder considered in the analysis. It also provides a comparison to PA 12.

In this table, the MFI values decrease slightly and continuously up to the 5th build cycle with a percentage difference of about 4.18% between the value of the virgin material and the powder sampled at the 5th build. After this, the values increase at higher rates with each powder re-use cycle, as illustrated in Figure 8. There is a 30.62% difference between the MFI value for the powder after the 9th build and that of the virgin material.

Polypropylene material behaves differently from PA 12. The latter shows a continuous reduction in MFI with powder re-use cycles, as described by Gornet et al. [19]. As noted above, the values of MFI of Laser PP CP 75 powder decrease at first up to the 5th re-use cycle and then start to increase with each re-use build. The findings up to the fifth build cycle confirm the study by Wegner & Ünlü [20], indicating that PP powder experiences a slight deterioration of rheological properties when used in the L-PBF process. The authors found that the MFI of PP decreases by only 3% when the powder is re-used for a single cycle. Changes giving rise to this trend are attributed to changes in the molecular weight of the materials, which also affects the degree of crystallinity of the powder. Gornet et al. [19] investigated the effects of repeated re-use of PA 12 from 3D Systems. The authors measured the flow characteristics using an extrusion plastometer, where it was established that MFR decreases with each build cycle, as indicated by Figure 9.

Kleijnen et al. [21] also found that the viscosity of PA 12 increases with processing time. The MFR of the powder used by the authors reduced significantly, from 28 cm^3^/10 min to 7 cm^3^/10 min after 4 h of storage in an oven at 170 °C. Aldahsh [22] established that the viscosity of a cement/PA 12 composite increases with processing time and temperature. It is evident from the preceding reviews that the characteristics of PA 12 change significantly with re-use cycles. The changes are much more significant than PP, as illustrated in Table 4. Hence, PP has superior characteristics compared to PA 12 in terms of the deterioration of rheological properties.

#### 4.2.2. Results of DSC for Laser PP CP 75 after 100% Re-Use

The DSC results were used to establish the melting point and sintering window of the various powder batches, each from a different number of recycles. Table 5 and Table 6 show the melting points and sintering windows of the five batches. Equations (1) and (2) were used to calculate the sintering window and crystallinity [23].

The sintering window (*SW*) is given by:(1)$$SW={\left(T_{m}\;–\;T_{c}\right)}_{\mathrm{onset}\;}$$ where
*T_m_* = onset melting point;*T_c_* = onset crystallization.


The degree of crystallization is given by:(2)$$DC=\left(∆\mathrm{Hm}/∆H^{0}{}_{m}\right)\times 100\%$$ where
DC = degree of crystallization;∆Hm = experimental melting enthalpy;∆H^0^m = theoretical melting enthalpy of the material, equal to 209 J/g [24].


In Figure 10, the melting points of Laser PP CP 75 increase with each re-use cycle, starting from a value of 134.48 °C after the 1st print cycle to 136.16 °C after the 4th re-use cycle. A new DSC testing device and software (DSC 2500) were used after the 4th re-use cycle. The results (5th–8th re-use cycles) varied from the previous readings (virgin–4th re-use cycles) and thus introduced an inconsistency that limited comparison. However, it was evident that the melting point increased from 132.63 °C to 133.06 °C after the 5th and 9th re-use cycles, respectively. The behavior of the material for the first four print cycles is similar to that of PA 12. Gornet et al. [19] used DSC to study the impact of re-using PA 12 on the material’s melting point. It was found that the melting point increases with the number of printing cycles, as illustrated in Figure 11. This trend of the melting point indicates that increasingly higher laser energy is required to fuse the material with each re-use cycle, which would contribute to further deterioration of the material. Suitable polymers should have a narrow melting point region to prevent the use of high laser energy when fusing powder particles. This is to avoid more significant degradation of the powder, supporting the components being printed and extending the recyclability of the powder [12,13]. However, as observed here, the melting point of Laser PP CP 75 increases with the number of re-use cycles, which necessitates a higher laser beam energy, encouraging further deterioration of the material.

The DSC results obtained here further illustrate that the sintering window of Laser PP CP 75 increases with each re-use cycle (Figure 12). This phenomenon might be attributed to the degradation and cross-linking of the long carbon chains [16]. A similar phenomenon was also observed by Dadbakhsh et al. [25], who investigated the effects of re-use cycles on the sintering window of PA 12. Figure 13 shows that the gap between melting and crystallization points is wider for aged powder than for virgin PA 12.

The sintering window of the new Laser PP CP 75, ranging from 18.14–25.18 °C. (Table 6), is lower than the sintering window for PA 12, which ranges from 32–34 °C [13]. Therefore, it is expected that difficulties would be experienced when regulating the cooling rate of the printed parts using the new Laser PP CP 75. It also explains the problems of curling observed in the present work while printing with the material. The issue of curling can be resolved by establishing the most suitable extraction and building chamber temperatures. Schmid et al. [26] state that homogeneous and stable thermal conditions minimize the curling and warpage phenomena. Their findings further indicate that the shrinkage and curling rates might reduce with the re-use cycles because of a wide and sufficient sintering window, preventing the crystallization of the polymers during processing [12,13]. Rapid crystallization of printed components is a significant hurdle in L-PBF because it encourages curling and affects the surface finish and dimensional accuracy of the parts [23]. The sintering window affects the cooling rate of printed parts, and materials with large sintering windows are known to exhibit even cooling rates. Hence, a larger sintering window decreases the curling and shrinkages rates of printed parts and vice versa for a lower sintering window. This makes the new Laser PP CP 75 powder less suitable than PA 12 powder in this respect.

Table 7 and Figure 14 illustrate the changes in the degree of crystallization of Laser PP CP 75 with re-use cycles.

The degree of crystallization increases up to a maximum point after the 3rd print cycle (2nd re-use cycle) and then starts to decrease. The findings concur with Dadbakhsh et al. [25], who found that the crystallinity of PA 12 increases with aging to a maximum point, after which it starts to decrease. The phenomenon might be explained through changes in the molecular weight of the powder. However, this was not within the scope of the present work.

#### 4.2.3. Results of TGA for Laser PP CP 75 after 100% Re-Use

The results of TGA analysis were used to establish the degradation and breakdown temperatures of the five batches of powder, each after several re-use cycles. The results of the tests are summarized in Table 8.

The breakdown temperature of Laser PP CP 75 increased slightly with the re-use cycles from 455.53 °C (virgin material) to 457.53 °C after the 4th re-use cycle (Figure 15), probably due to increased crystallinity of the material from cross-linking of the long carbon chains. Building and removal chamber temperatures of 125 °C and 128 °C were used when printing the material. These temperatures are significantly lower than the registered breakdown temperatures in Table 8. Hence, the material is not expected to break down during printing, making it suitable for L-PBF processing. Marin [13] stated that polymers should have high degradation temperatures because L-PBF occurs at high temperatures.

#### 4.2.4. Results of SEM for Laser PP CP 75 after 100% Re-Use

Scanning electron microscopy was undertaken to determine changes of morphology and powder particle distribution of the PP powder after each re-use cycle. Table 9 illustrates the distribution of particle sizes of Laser PP CP 75 for fresh material and after different re-use cycles. The particle size distribution was determined using ImageJ software (1.53n/7).

The powder particle size distribution for Laser PP CP 75 is far above the recommended range, which should be between 20 µm and 80 µm, according to Schmid and Wegener [27], and 45 and 90 μm based on a study by Schmidt et al. [28]. It was noted in the literature review that extremely large particles affect the spreading of powder using a recoater roller or blade. Large particles also discourage fusion, which introduces porosity and, in turn, reduces the mechanical integrity of printed parts [29]. The mean particle size varied for the nine batches of powder. The significant mean particle size difference was between the third and seventh re-use cycles, with a percentage difference of 42.93%. This change can be attributed to the re-use of the powder. Dadbakhsh et al. [25] state that the particle size of polymeric powder particles is not significantly influenced by re-use cycles. The data in Table 9 shows that the build layer thickness for processing Laser PP CP 75 should be around 300 µm because, according to Berretta et al. [30], the build layer thickness for processing should be at least twice the average size of the powder particles. This ensures that powder fusion happens in direct contact with the laser beam rather than having particle-to-particle conduction. Particle-to-particle conduction results in the partial coalescence of the particles that then leads to the production of mechanically weak components.

**Figure 13 polymers-14-01011-f013:**
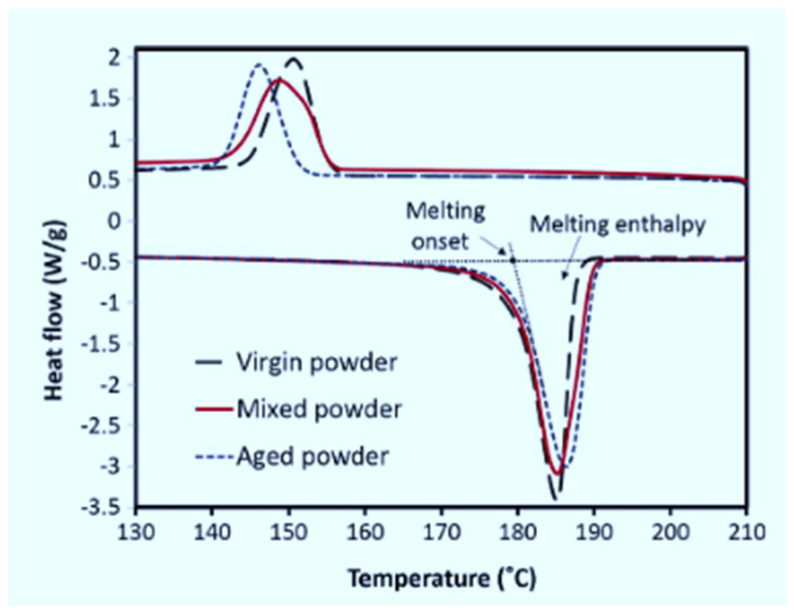
DSC thermogram for virgin, aged and mixed PA 12 powder [28].

**Figure 14 polymers-14-01011-f014:**
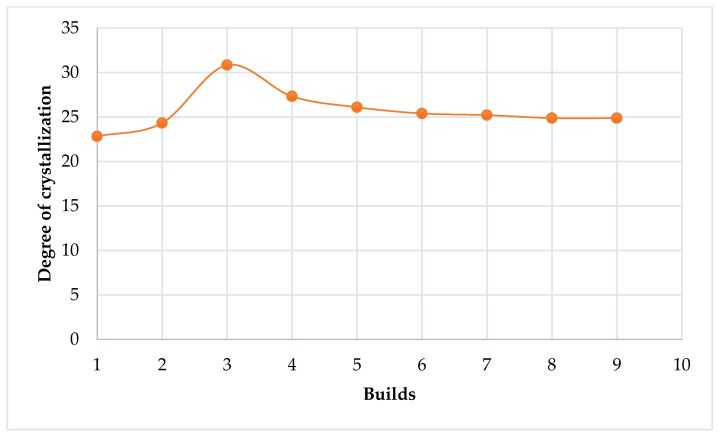
Degree of crystallization of Laser PP CP 75 versus re-use cycles.

**Figure 15 polymers-14-01011-f015:**
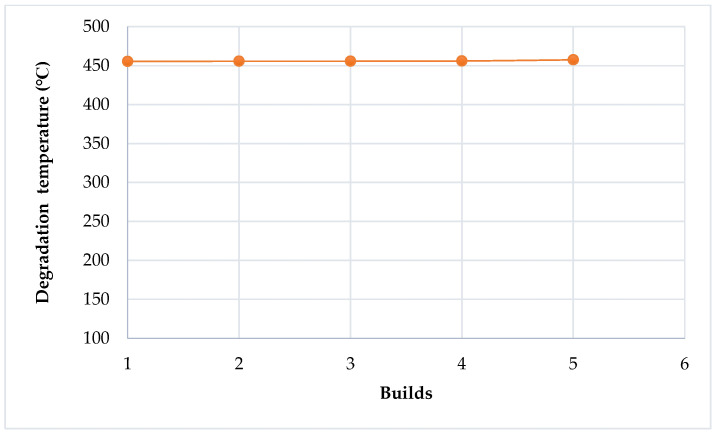
Degradation temperature of Laser PP CP 75 versus re-use cycles.

The effect of recycling on the morphology and agglomeration of Laser PP CP 75 was investigated using SEM, and the micrographs obtained are shown in Figure 16. The micrographs show that Laser PP CP 75 is a composite of two materials. One of the materials has irregularly shaped particles, and the other material has perfectly spherical particles, both of different sizes. The irregularly shaped particles are Laser PP CP 75 polymer particles consisting of pure PP. The perfectly shaped particles are glass beads (according to the manufacturer) that were added to improve the difficulties of flowability experienced with Laser PP CP 60. The images in Figure A2 illustrate that Laser PP CP 75 does not experience significant agglomeration with the re-use cycles.

### 4.3. Part Characterization to Establish Recyclability of Laser PP CP 75

After each re-use cycle, the printed parts were also subjected to tensile testing and analysis for dimensional accuracy to investigate changes in the mechanical and physical properties of the components printed using Laser PP CP 75 material.

#### 4.3.1. Uniaxial Tensile Test for Laser PP CP 75 after 100% Re-Use

The average values of tensile strength, Young’s modulus, and the percentage elongation point for the five samples tested after each re-use cycle are summarized in Table 10, Table 11 and Table 12, respectively. The trends of these mechanical properties are illustrated in Figure 17, Figure 18, and Figure 19.

In Figure 17, the tensile strength of the printed parts increased slightly, from 6.7 MPa to 7.4 MPa, with the re-use of powder up to the 2nd re-use cycle. After the 2nd re-use cycle, the tensile strength value of the printed parts decreases to below that of the parts printed from fresh powder.

The Young’s modulus of the printed parts decreases continuously with the re-using of the powder up to the 2nd re-use cycle from an initial value of 1141.273 MPa for the 1st print cycle to 807.638 MPa after the 3rd print cycle. After this, Young’s modulus starts to increase with each re-use cycle.

The percentage elongation of the printed parts decreases continuously with the re-use of powder up to the 2nd re-use cycle, starting with 61.91% for parts printed with virgin powder to 29.73% after the 3rd print cycle. After the 2nd re-use cycle, the percentage elongation increases for the next re-use cycle and then levels out beyond this.

The trends seen in the curves of the variation of the strength, stiffness, and percentage elongation of printed parts might be due to changes in the crystallinity of the material (see Figure 14). This is due to the entanglement and ordered chain folding of the long carbon chains during the L-PBF process, which rise and drop with increased crystallinity, respectively [25]. Therefore, increased crystallinity will raise the ultimate tensile strength of printed parts at the expense of reduced stiffness and percentage elongation to break [12]. It can be seen from the preceding figures that Laser PP CP 75 attained the highest ultimate tensile strengths in the 2nd re-use cycle.

Another study by Wegner & Ünlü [20] concluded that the mechanical characteristics of parts produced using PP reduce with the number of re-use cycles (Figure 20). The reduction in mechanical properties is subject to the energy density of the laser beam used.

#### 4.3.2. Dimensional Accuracy of Parts Printed Using Laser PP CP 75

The dimensional accuracy of the printed parts was expressed using error *S*_1_, as recorded in Table 13. The dimensional error *S*_1_ was calculated using Equation (3) by Singh et al. [31]:(3)$$S_{1}=\frac{A_{1}-A_{0}}{A_{0}}*\;100\%$$ where
*A*_0_ = Design size of the CAD diagram;*A*_1_ = Actual size measured using a Vernier caliper.


Table 13 indicates that the dimensions of the printed parts differ from those of the CAD diagram, with maximum errors of up to 10%, 0.68%, and 6.5% for the thickness, length, and width occurring for the 5th print cycle parts, respectively. The errors are thought to have been caused by the expansion of parts due to the effect of heating during the L-PBF process [31]. The trend for the average dimensional error is presented in Figure 21 and was used as a representative figure for ease of analysis. The average dimensional error increased for the 1st re-use cycle from 3.02% to 4.78%. It then dropped to 1.33% for the 3rd print cycle. After this, the average dimensional error increased to 2.87% and 4.06% for the 4th and 5th print cycles, respectively. The lowest average dimensional error of 1.33% was achieved from parts printed during the 3rd print cycle. Overall, the dimensional variation of the printed parts indicates that Laser PP CP 75 material might not be suitable for commercial purposes due to the high dimensional tolerance required for commercial parts. Besides, significant curling was experienced, which affected the edges of the printed components.

## 5. Conclusions

It was concluded that PP powder (Laser PP CP 75) from Diamond Plastics GmbH could be re-used 100% for five print cycles without mixing with virgin material because it does not form an “orange peel”, as is the case with PA 12 (used in this study as reference material). After this cycle, the printed parts have distorted edges. The MFI values of the powder exhibited small changes, with a 4.1% decrease after the 4th re-use cycle, indicating that the viscosity of the material does not degrade significantly due to re-use and therefore promotes recyclability of the powder. The melting point increased from 134.48 °C to 136.16 °C after the first four re-use cycles, which indicates that higher laser energy might be required to fuse the powder with increasing re-use. The DSC test results established that the sintering window of Laser PP CP 75 increased with each re-use cycle. Based on the TGA test, the breakdown temperature of Laser PP CP 75 increased slightly with the number of re-use cycles, from 455.53 °C (virgin material) to 457.53 °C after the 4th re-use cycle. The SEM analysis revealed that Laser PP CP 75 is a composite with irregularly shaped particles, which might impede spreading, but the powder did not exhibit signs of agglomeration for any of the re-use cycles. The spherical particles in the powder were assumed to be flow agents (glass beads). Tensile testing revealed that the parts built from this powder attained the highest ultimate tensile strength, lowest stiffness, and percentage elongation after the 3rd print cycle (2nd recycle cycle) (7.4 MPa), after which these values decreased and increased with each recycling, apart from percentage elongation that increased first and then leveled out. The parts printed with virgin Laser PP CP 75 had an average dimensional error of 3.02% for the virgin powder and 4.06% after the 4th re-use cycle, which is not significant but nonetheless important to note where high dimensional tolerance is required. Therefore, it is safe to conclude that Laser PP CP 75 may not be suitable for commercial applications because of the observed significant errors of dimensional accuracy.

## 6. Recommendation

Further experiments should be conducted to optimize the process parameters of Laser PP CP 75 to determine if the material can be used to print complex parts.

## Figures and Tables

**Figure 1 polymers-14-01011-f001:**
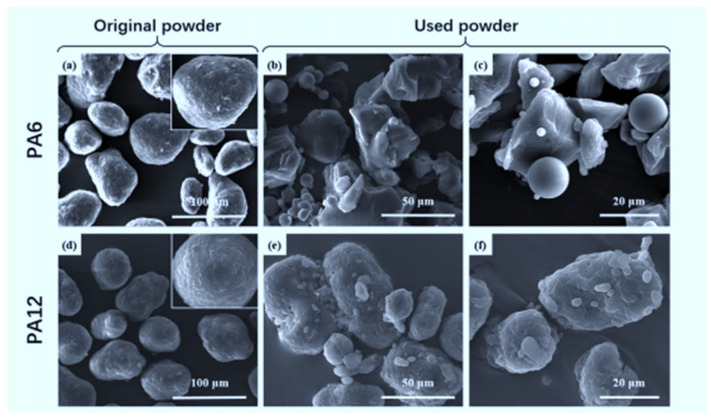
Differences in the morphology of original and used PA 6 and PA 12 powders. Reprinted with permission from ref. [14] copyright 2018 Elsevier. SEM images of (**a**) PA6, Original powder (100 μm), (**b**) PA6, Used powder (50 μm), (**c**) PA6, Used powder (20 μm), (**d**) PA12, Original powder (100 μm), (**e**) PA12, Used powder (50 μm), (**f**) PA12, Used powder (20 μm).

**Figure 2 polymers-14-01011-f002:**
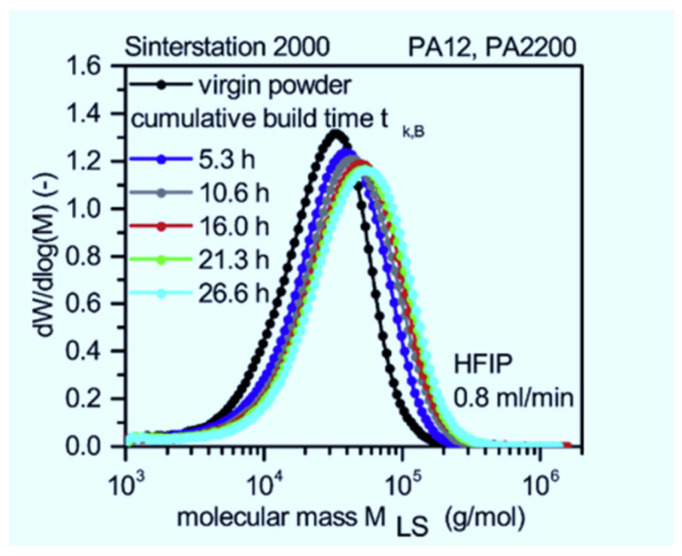
Effects of cumulative build time on the average molecular weight of PA 12 [16].

**Figure 3 polymers-14-01011-f003:**
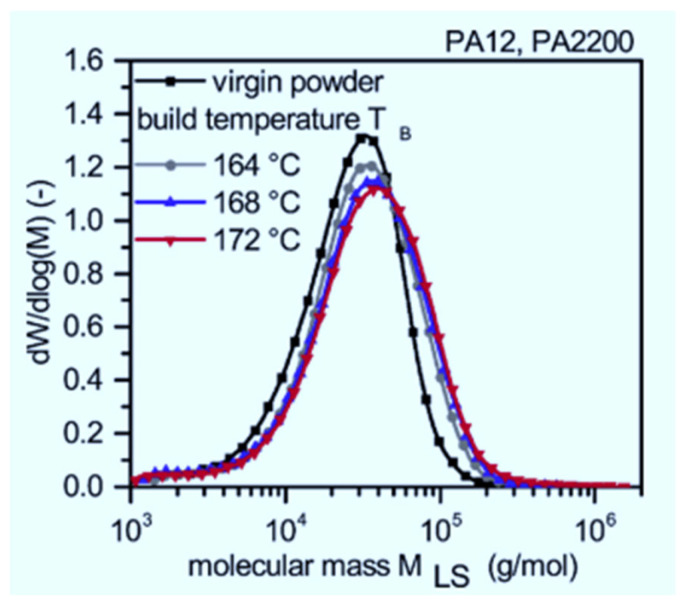
Effects of build chamber temperature on the average molecular weight of PA 12 [16].

**Figure 4 polymers-14-01011-f004:**
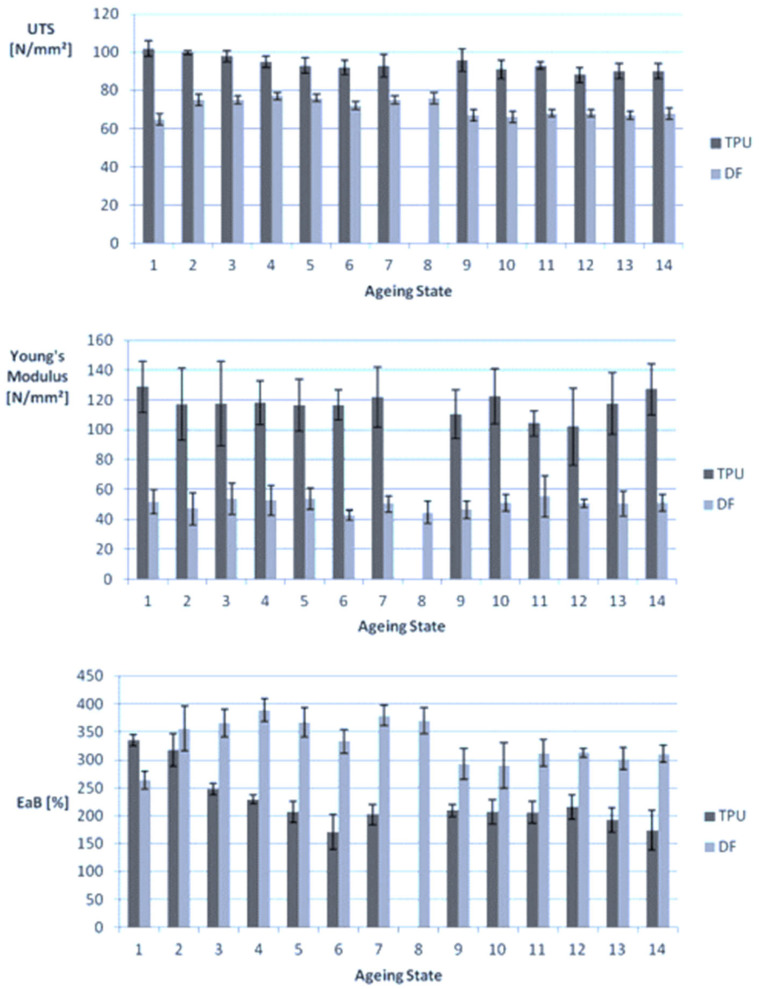
Changes in mechanical properties of parts printed using TPU and DF over different aging states.

**Figure 5 polymers-14-01011-f005:**
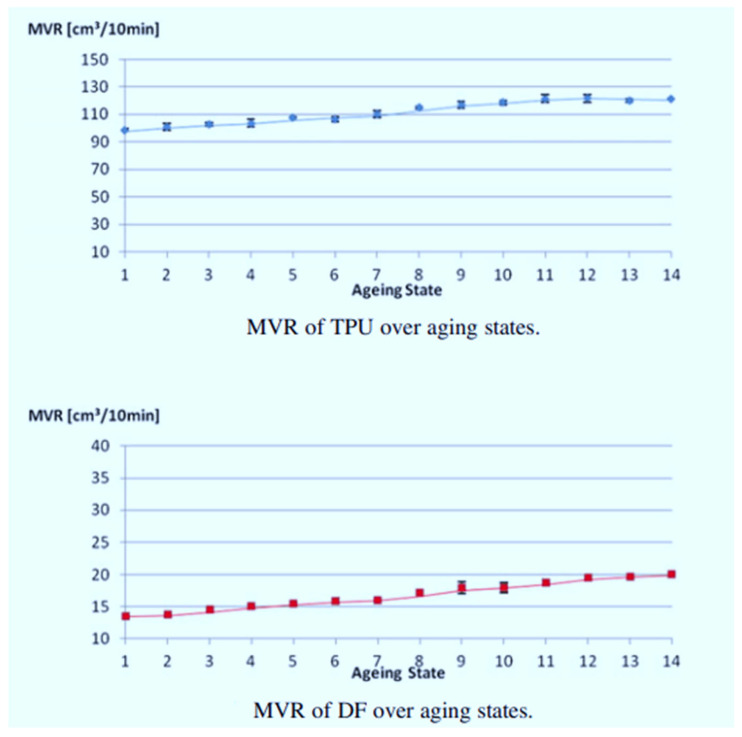
Changes in melt volume-flow rate for TPU and DF over different aging states [17].

**Figure 6 polymers-14-01011-f006:**
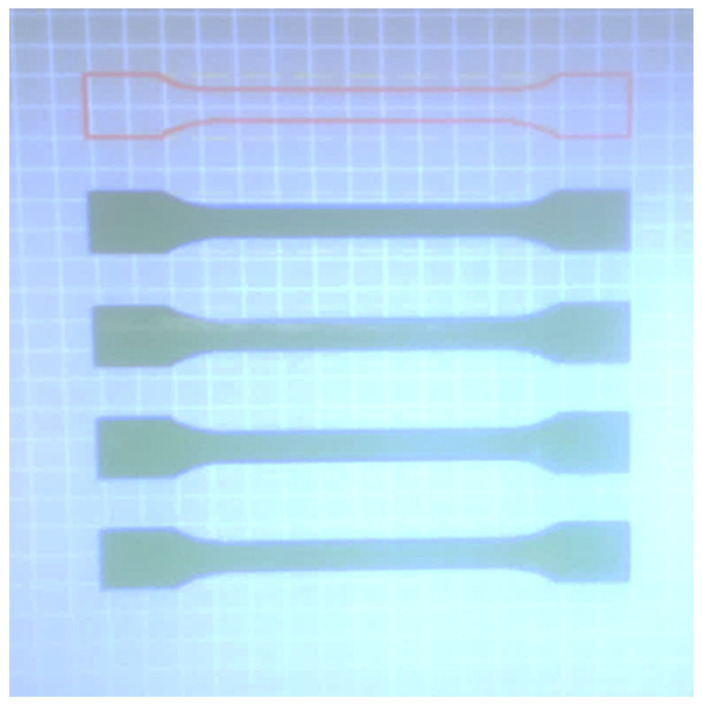
The set of test coupons at predetermined positions.

**Figure 7 polymers-14-01011-f007:**
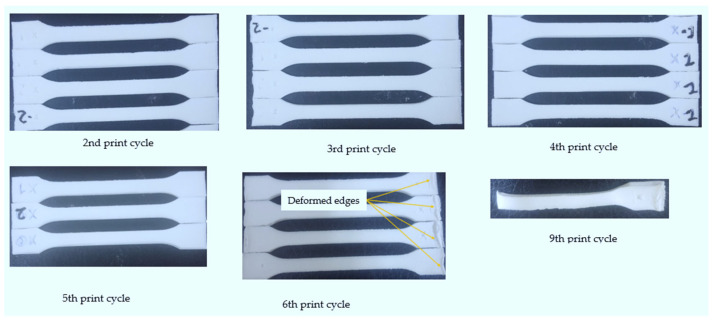
Printed tensile test specimens for different re-use powder cycles.

**Figure 8 polymers-14-01011-f008:**
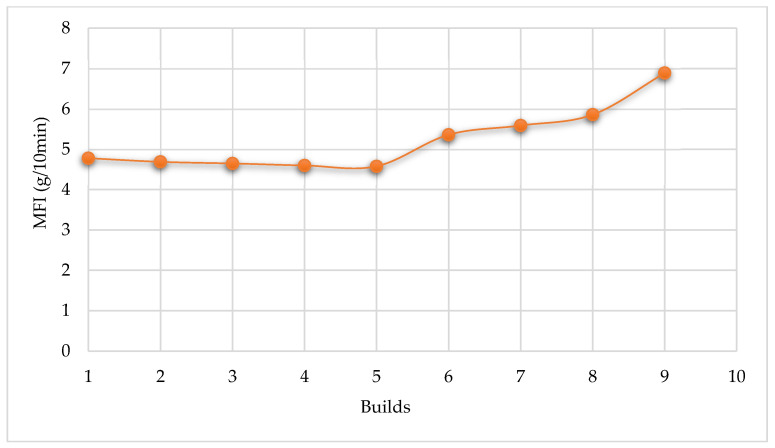
The trend of the MFI values up to nine builds.

**Figure 9 polymers-14-01011-f009:**
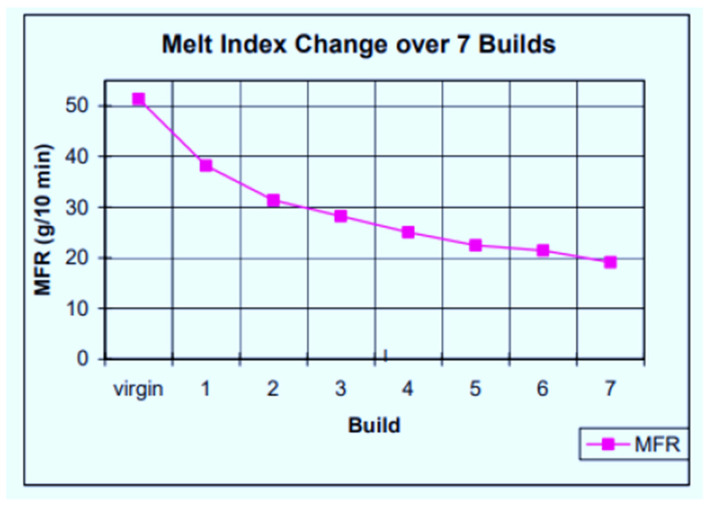
MFR of PA 12 over seven builds [19].

**Figure 10 polymers-14-01011-f010:**
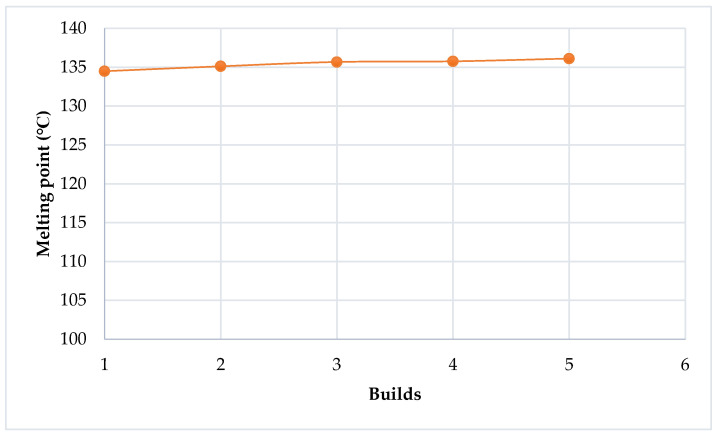
Melting points of Laser PP CP 75 over a number of re-use cycles.

**Figure 11 polymers-14-01011-f011:**
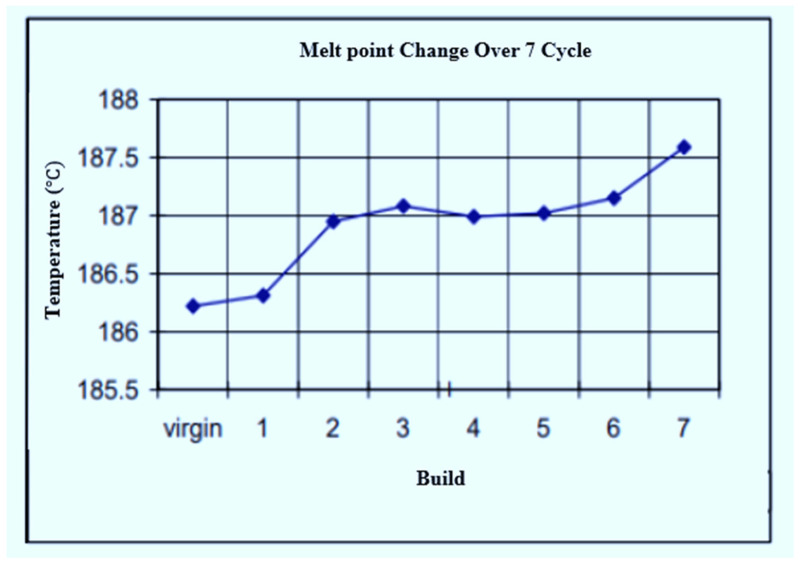
Melting points of PA 12 over seven re-use cycles [19].

**Figure 12 polymers-14-01011-f012:**
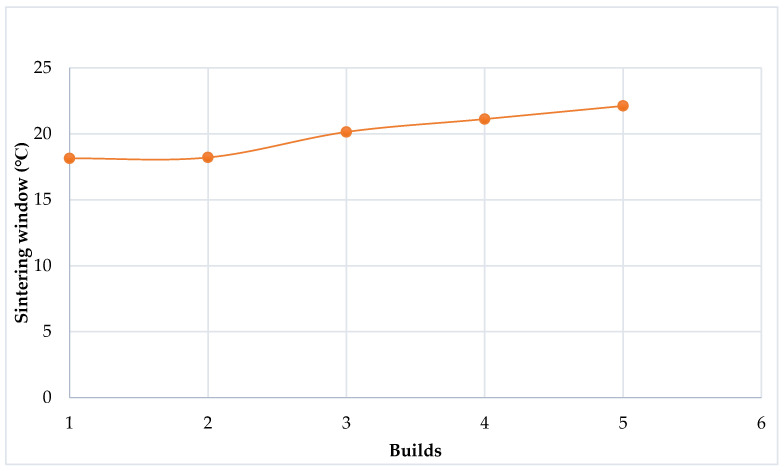
Sintering window of Laser PP CP 75 versus re-use cycles.

**Figure 16 polymers-14-01011-f016:**
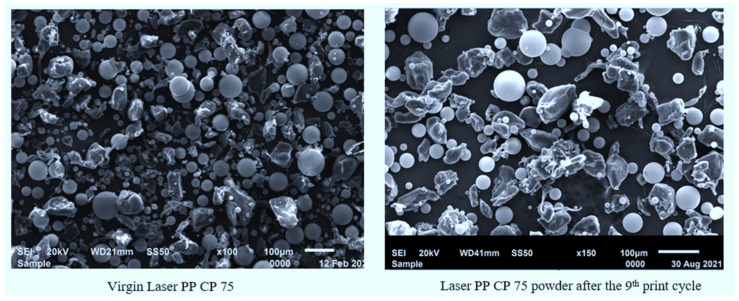
Powder morphology for Laser PP CP 75.

**Figure 17 polymers-14-01011-f017:**
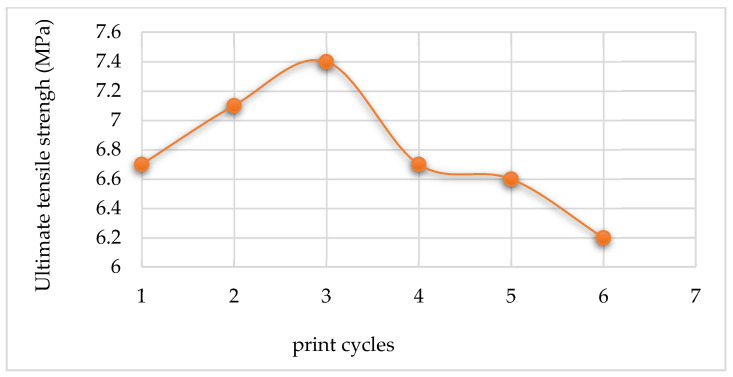
Ultimate tensile strengths of parts printed using Laser PP CP 75.

**Figure 18 polymers-14-01011-f018:**
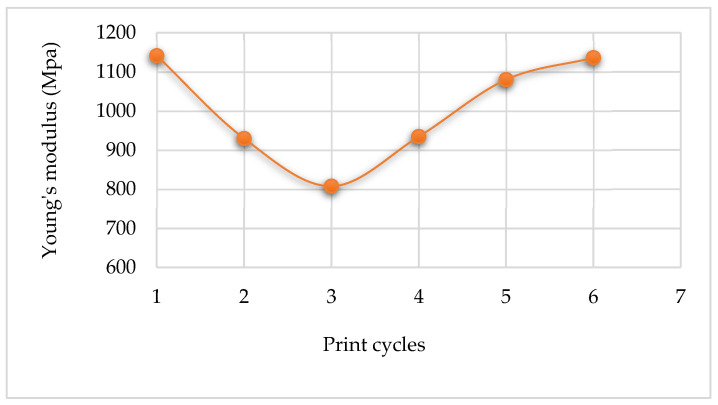
Young’s modulus of parts printed using Laser PP CP 75.

**Figure 19 polymers-14-01011-f019:**
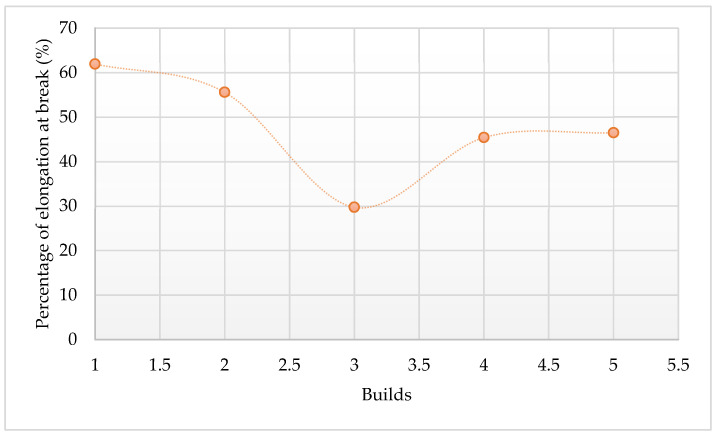
Percentage elongation at break of Laser PP CP 75 printed parts.

**Figure 20 polymers-14-01011-f020:**
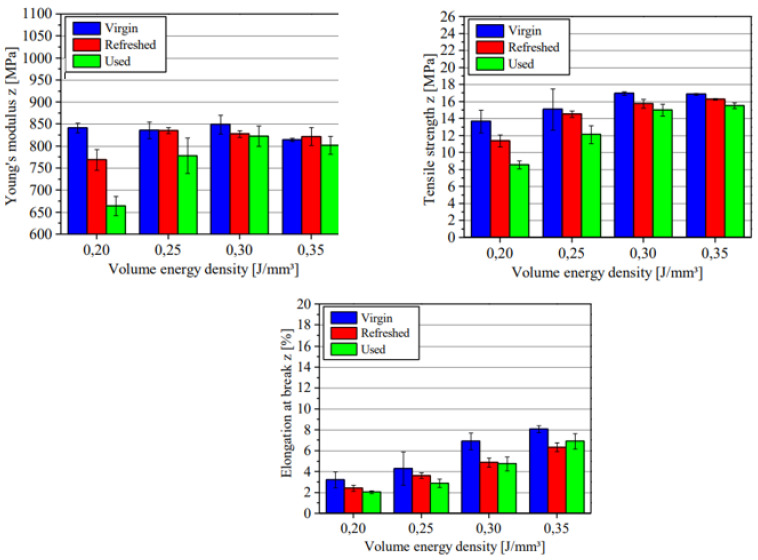
Effects of re-use cycles on the mechanical properties of PP [20].

**Figure 21 polymers-14-01011-f021:**
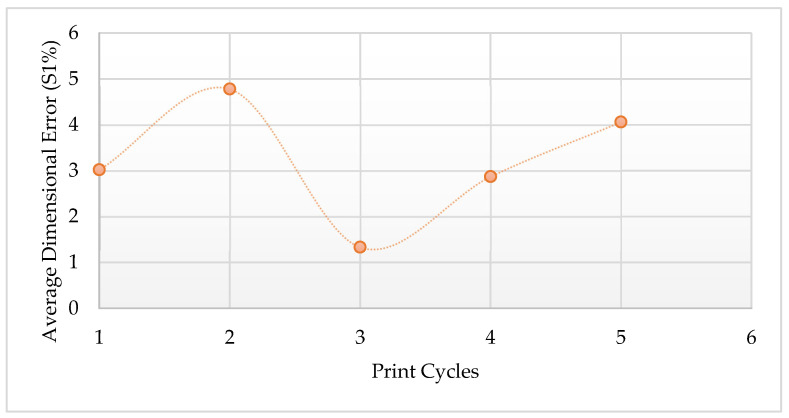
Average dimensional error for parts printed using Laser PP CP 75.

**Table 1 polymers-14-01011-t001:** Particle size distribution of the two powders after a single print cycle [14].

Data	Original PA 6 Powder	Used PA 6 Powder	Original PA 12 Powder	Used PA 12 Powder
D_V_ (10)/μm	43.1	23.4	34.4	33.2
D_V_ (50)/μm	58.9	52.0	53.8	52.9
D_V_ (90)/μm	80.3	91.2	80.9	80.2

**Table 2 polymers-14-01011-t002:** Changes of thermal properties of PA 6 and PA 12 after a single print cycle [14].

Materials		Melting Enthalpy (J/g)	Crystallinity (%)	Sintering Window (°C)
PA 6	Original powder	109.50 ± 1.04	47.61 ± 0.45	36.73 ± 0.02
	Used powder	50.14 ± 2.75	21.80 ± 1.20	23.58 ± 0.045
PA 12	Original powder	98.24 ± 0.18	46.94 ± 0.08	30.65 ± 0.01
	Used powder	92.56 ± 1.31	44.22 ± 0.63	30.86 ± 0.02

**Table 3 polymers-14-01011-t003:** Parameters for recyclability of Laser PP CP 75.

Temp. of the Removal Chamber (°C)	Temp. of the Building Bed (°C)	Layer Thickness (mm)	Hatch Distance (mm)	Scanning Speed Fill (mm/s)	Laser Power Fill (W)	Scanning Speed Contour/Edges (mm/s)	Laser Power Contour/Edges (W)
125	128	0.15	0.25	4500	35.0	1500	20.0

**Table 4 polymers-14-01011-t004:** MFI values for the nine batches of powder considered in the analysis.

	Powder Batch	Value of MFI (g/10 min) of Laser PP CP 75	Percentage Change in Virgin Material	Value of MFI (g/10 min) for PA 12 from a Study by Gornet et al. [15]	Percentage Change in Virgin Material
#	Manufacturer’s stated value	4.50	-	-	-
1	Virgin material	4.78	0.00%	53	0.00%
2	Re-used powder after 1st cycle	4.69	1.88%	38	28.30%
3	Re-used powder after 2nd cycle	4.65	2.72%	32	39.62%
4	Re-used powder after 3rd cycle	4.60	3.77%	27	49.06%
5	Re-used powder after 4th cycle	4.58	4.18%	25	52.83%
6	Re-used powder after 5th cycle	5.36	10.82%	22	58.49%
7	Re-used powder after 6th cycle	5.59	14.49%	18	66.04%
8	Re-used powder after 7th cycle	5.86	18.43%	-	-
9	Re-used powder after 8th cycle	6.89	30.62%	-	-

**Table 5 polymers-14-01011-t005:** Changes in melting point of Laser PP CP 75 after eight re-use cycles.

Powder Batch	Peak Melting Point °C
Fresh material	134.48
1st re-use cycle	135.11
2nd re-use cycle	135.66
3rd re-use cycle	135.75
4th re-use cycle	136.10
5th re-use cycle	132.63
6th re-use cycle	132.78
7th re-use cycle	132.96
8th re-use cycle	133.06

**Table 6 polymers-14-01011-t006:** Calculated sintering window of Laser PP CP 75 after four re-use cycles.

Powder Batch	Sintering Window, SW (°C)
Fresh material	18.14
1st re-use cycle	18.21
2nd re-use cycle	20.14
3rd re-use cycle	21.12
4th re-use cycle	22.12
5th re-use cycle	21.40
6th re-use cycle	21.43
7th re-use cycle	25.03
8th re-use cycle	25.18

**Table 7 polymers-14-01011-t007:** Changes of a degree of crystallization of Laser PP CP 75 after eight re-use cycles.

Powder Batch	Degree of Crystallization (%)
Fresh material	22.83
1st re-use cycle	24.34
2nd re-use cycle	30.85
3rd re-use cycle	27.33
4th re-use cycle	26.11
5th re-use cycle	25.41
6th re-use cycle	25.22
7th re-use cycle	24.89
8th re-use cycle	24.88

**Table 8 polymers-14-01011-t008:** TGA results for Laser PP CP 75 after 100% re-use.

Powder Batch	Degradation Temperature (°C)
Fresh material	455.53
1st re-use cycle	455.75
2nd re-use cycle	455.75
3rd re-use cycle	456.04
4th re-use cycle	457.53

**Table 9 polymers-14-01011-t009:** Particle size distribution.

Powder Batch	Powder Particle Size Distribution (µm)	Mean Powder Particle Size (µm)	Standard Deviation of the Particle Size (µm)
Fresh material	64.54–248.49	150.23	39.30
1st re-use cycle	50.45–192.68	123.87	32.00
2nd re-use cycle	74.73–266.59	148.18	39.40
3rd re-use cycle	59.54–273.94	158.66	43.43
4th re-use cycle	69.44–274.65	156.32	36.45
5th re-use cycle	47.27–192.74	95.02	27.26
6th re-use cycle	99.78–202.96	125.34	24.07
7th re-use cycle	49.62–147.06	90.54	26.38
8th re-use cycle	71.72–154.39	117.29	17.41

**Table 10 polymers-14-01011-t010:** Ultimate tensile strength of printed parts after 100% re-use.

#	Powder Batch	Ultimate Tensile Strength (MPa)
1	Virgin powder (1st print cycle)	6.7
2	Re-used powder (2nd print cycle)	7.1
3	Re-used powder (3rd print cycle)	7.4
4	Re-used powder (4th print cycle)	6.7
5	Re-used powder (5th print cycle)	6.6
6	Re-used powder (6th print cycle)	6.2

**Table 11 polymers-14-01011-t011:** Young’s modulus of printed parts after 100% re-use.

#	Powder Batch	Young’s Modulus (MPa)
1	Virgin powder (1st print cycle)	1141.273
2	Re-used powder (2nd print cycle)	929.623
3	Re-used powder (3rd print cycle)	807.638
4	Re-used powder (4th print cycle)	934.513
5	Re-used powder (5th print cycle)	1080.075
6	Re-used powder (6th print cycle)	1136.033

**Table 12 polymers-14-01011-t012:** Percentage of elongation at break of printed parts after 100% re-use.

#	Powder Batch	Percentage of Elongation at Break (%)
1	Virgin powder (1st cycle)	61.91
2	Re-used powder (2nd cycle)	55.56
3	Re-used powder (3rd cycle)	29.73
4	Re-used powder (4th cycle)	45.40
5	Re-used powder (5th cycle)	46.46

**Table 13 polymers-14-01011-t013:** Dimensional errors of L-PBF printed parts.

Measurement	Specimen 1 (S1%)	Specimen 2 (S1%)	Specimen 3 (S1%)	Specimen 4 (S1%)	Specimen 5 (S1%)
Thickness	5.0	7.5	2.5	5.0	10.0
Length	0.55	0.34	0.0	0.62	0.68
Width	3.5	6.5	1.5	3.0	1.5
Average (S1%)	3.02	4.78	1.33	2.87	4.06

Specimen 1—Parts produced using virgin material (1st cycle). Specimen 2—Parts produced using re-used powder (2nd cycle). Specimen 3—Parts produced using re-used powder (3rd cycle). Specimen 4—Parts produced using re-used powder (4th cycle). Specimen 5—Parts produced using re-used powder (5th cycle).

## Data Availability

The data presented in this study are available on request from the corresponding author.

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
