# Peer review of "Investigating the Recyclability of Laser PP CP 75 Polypropylene Powder in Laser Powder Bed Fusion (L-PBF)"

_polymers, 2022, doi:10.3390/polym14051011_

Round 1
Reviewer 1 Report
The authors investigated the recyclability of laser PP CP 75 material for LPBF technologies.
Innovative aspects of the article;
- The results are presented in comparison with the most commonly used PA12 in this technology.
- An innovative and detailed study on the recyclability of PP75 material.
- Characterization techniques are adequate and detailed, comments are given in detail.
- The language of the article is simple and the writing is fluent.
I think the article can be accepted after some minor corrections;
- Information on the use of PP CP 75 material should be added to the introduction section (where this material is used, what is its importance)
- In the introduction, many examples of polymer-based materials are given. A brief addition with Primecast, a special polymer used from SLS techniques, will enrich the article (see this link on this subject: https://www.tandfonline.com/doi/full/10.1080/13640461.2020.1773053)
Author Response
- Information on the use of PP CP 75 material should be added to the introduction section (where this material is used, what is its importance) - this information was added in the introduction given in lines 52 and 55.
- In the introduction, many examples of polymer-based materials are given. A brief addition with Primecast, a special polymer used from SLS techniques, will enrich the article - this information was added to the introduction.
Reviewer 2 Report
Fredrick M. Mwania et al have investigated the recyclability of laser PP CP 75 polypropylene powder in powder bed fusion technique. The manuscript is well written and the resued powder batches are characterized satisfactorily. Hence, I recommend this manuscript for publication in Polymer journal if the author addresses the following comments.
-
The abstract is too long, it should be more concise.
-
There are many grammatical errors and spelling mistakes in the manuscript e.g "molecualr weigh. The author should revise this manuscript carefully.
-
Figure 16, does not give a scale bar. Two figures should be in the same dimensions.
-
Why the Young modulus has increased after the 4th cycle in Table 12?
-
All the figures and tables resolution needs to be increased
-
The conclusion looks too lengthy.
Author Response
-
The abstract is too long, it should be more concise - The Abstract was reduced to 218 words.
-
The manuscript has many grammatical errors and spelling mistakes, e.g., "molecular weight. The author should revise this manuscript carefully - The authors did another round of revision for grammar checks. The paper was also revised using Grammarly software.
-
Figure 16 does not give a scale bar. Two figures should be in the same dimensions - The scale bars have been provided.
-
Why the Young modulus has increased after the 4th cycle in Table 12 - The discussion on this matter is provided in lines 411 to 422.
-
All the figures and tables resolution needs to be increased - the pictures were enhanced using different photo enhancement tools on Word to improve their resolution.
-
The conclusion looks too lengthy - The conclusion was reduced to 352 words.
This manuscript is a resubmission of an earlier submission. The following is a list of the peer review reports and author responses from that submission.
Round 1
Reviewer 1 Report
This paper presented the recyclability of Laser PP CP 75 polypropylene powder from Diamond Plastics GmbH in laser powder bed fusion process. After print cycles Laser PP CP 75 was characterized by melt flow testing, DSC, TG, SEM and tensile testing. Although the results are meaningful, however, there are still left many unclear questions in the manuscript. The work in this paper is needed to be improved thoroughly. Specifically, there are some technical problems as listed below.
- The experimental information in detailon the Laser PP CP 75 polypropylene powder, such as the average molecular weight, molecular weight distribution, isotacticity, density, etc., should be provided.
- How is the formed PP sample processed into powder after each cycle?Authors should clearly describe the relevant experimental details.
- The authors mention a slight decrease in the MFI of PP due to the changes in the molecular weight of the materials and the degree of crystallinity of the PP powder was also affected. It is recommended to prove this through molecular weight test using GPC and crystallinity test by DSC. There is no crystallinity data in the DSC analysis in this paper.
- The important TG and DSC curves of the PP samples after printing should be presented for better understanding of the results.
- Some figures in the paper areblurred, so pictures with higher resolution should be provided.
- Figures 7 to 14 should be integrated into one figure for comparison.
- The scale barsin the SEM figures are not clear.
- Authors should compare Laser PP CP 75 with other commonly used commercial PP powders through experiments and tests besides PA 12.
- Authors had better summarizewhat factors for PP powder for laser powder bed fusion process play dominant roles for the better recyclability to improve readers’ understanding.
Reviewer 2 Report
The manuscript “Investigating the recyclability of laser PP CP 75 polypropylene powder in laser powder bed fusion (L-PBF)” a recyclability study of Laser polypropylene powder from Diamond Plastic GmbH was performed and compared with PA 12. The thermal and mechanical properties were analyzed after several recycled cycles. Please consider the following observations:
1.- After reading your document, your study is comparative between PP and PA6, perhaps you should make adjustments to your title.
2.- I consider that the bibliographic review part is not a typical review, since it includes tables and figures from other authors, these can be described or used in the analysis of their results.
3.- Improve methodology, include material data (PA 12?), better describe equipment (including make, model, city and country of manufacture).
4.- The next Figures an Tables have the same information: Fig 15 and Table 5, Fig 17 and Table 6, Fig 19 and Table 7, Fig 21 and Table 8, same for the Figure 23 and Table 10, Figure 24 and Table 11, Fig 25 and Table 12, Fig 27 and Table 13. Improve Table 3.
5.- Improve the conclusions, since it presents a summary of results, some are not outstanding
6.- Use journal abbreviations in your references